# Fall-from-Height Detection Using Deep Learning Based on IMU Sensor Data for Accident Prevention at Construction Sites

**DOI:** 10.3390/s22166107

**Published:** 2022-08-16

**Authors:** Seunghee Lee, Bummo Koo, Sumin Yang, Jongman Kim, Yejin Nam, Youngho Kim

**Affiliations:** Department of Biomedical Engineering, Yonsei University, Wonju 26493, Korea

**Keywords:** fall-from-height, IMU sensor, deep learning, risk prediction

## Abstract

Workers at construction sites are prone to fall-from-height (FFH) accidents. The severity of injury can be represented by the acceleration peak value. In the study, a risk prediction against FFH was made using IMU sensor data for accident prevention at construction sites. Fifteen general working movements (NF: non-fall), five low-hazard-fall movements, (LF), and five high-hazard-FFH movements (HF) were performed by twenty male subjects and a dummy. An IMU sensor was attached to the T7 position of the subject to measure the three-axis acceleration and angular velocity. The peak acceleration value, calculated from the IMU data, was 4 g or less in general work movements and 9 g or more in FFHs. Regression analysis was performed by applying various deep learning models, including 1D-CNN, 2D-CNN, LSTM, and Conv-LSTM, to the risk prediction, and then comparing them in terms of their mean absolute error (MAE) and mean squared error (MSE). The FFH risk level was estimated based on the predicted peak acceleration. The Conv-LSTM model trained by MAE showed the smallest error (MAE: 1.36 g), and the classification with the predicted peak acceleration showed the best accuracy (97.6%). This study successfully predicted the FFH risk levels and could be helpful to reduce fatal injuries at construction sites.

## 1. Introduction

Fall-from-height (FFH) accidents account for an extremely high proportion of accidents at construction sites with a fairly high mortality rate. Choi et al. [1] conducted a comparative analysis of accidents that occurred between 2011 and 2015 in three countries: the United States, Korea, and China. Accidents were found to occur frequently at construction sites, with the U.S. showing a 26% increase (from 781 to 985), while China and Korea showed a 28% decrease (2634 to 1891) and a 21% decrease (from 621 to 493), respectively. The average mortality rate was the highest in Korea (17.9 persons), followed by the U.S. and China (9.4 and 5.3 persons, respectively).

The Occupational Safety and Health Administration (OSHA) requires implementing physical safety measures to reduce such accidents at construction sites [2]. Primary protection measures include implementing guardrails, covers, safety nets, and physical safety devices, while secondary protection measures include the use of a personal fall arrest system (PFAS) whereby the impact of an FFH accident can be minimized [3]. A PFAS comprises a connector, full-body harness, lanyard, and rescue line, and may prevent a person from falling when properly configured [4]. A PFAS cannot prevent FFHs but can effectively avoid fatalities from FFHs [5]. Yang et al. [2] reported that fatalities due to losing balance can be avoided when the PFAS is properly used; however, if a worker is suspended from a PFAS for a prolonged time, there is a risk of suspension trauma, orthostatic intolerance, or other serious injuries, and workers still sustain injuries due to the incompleteness of a PFAS [6]. Furthermore, the effect of a PFAS is insignificant when a person falls from a height below 15 ft, and accidents occur because workers do not properly wear PFAS due to their inconvenience during work [7].

To address the limitations of interrupting the movements of workers, some studies have considered the use of visual devices [8,9,10]. Han et al. [8] analyzed four actions on ladders using depth information measured by Kinect and classified unsafe actions with 90.9% accuracy. Fang et al. [9] defined unsafe behaviors as situations in which workers and the structural supports overlap in construction. They developed an automatic computer-vision system with 90% recall and 75% precision using CNN. Kong et al. [10] combined computer vision with long short-term memory (LSTM) to predict unsafe actions from video data. They determined the safety of actions based on the predicted trajectories. Their model showed a mean intersection-over-union of 73.4%, mean absolute precision of 92.9% (IOU: 0.5), and mean absolute precision of 68.1% (IOU: 0.7). However, the vision-based system is not effective since workers might be obscured by structures on construction sites.

Researchers have actively developed fall detection algorithms to minimize injuries in the elderly from falls using wearable sensors [11,12,13]. Threshold-based methods have mainly been used for fall detection. Jung et al. [11] developed a fall detection algorithm with an accuracy of 92.4% and a lead time of 280.25 ± 10.29 ms, evaluated on the SisFall public dataset, and used a complementary filter to compute the vertical angles from the IMU sensor data. Ahn et al. [12] developed a hip protection system for the elderly, which comprises an IMU sensor, a non-gunpowder type inflator, and a wearable airbag with a threshold-based fall detection algorithm, and 100% accuracy and a 401.9 ± 46.9 ms lead time were obtained. Koo et al. [13] developed a post-fall detection algorithm based on machine learning using an IMU sensor. They used five different ranking algorithms to select feature subsets. The feature subsets selected by the T-score showed the best accuracy of 99.86%.

Several studies have been conducted on developing FFH detection algorithms by extending the aforementioned research [2,14,15]. Yang, et al. [2] performed near-miss fall detection based on machine learning with IMU sensor data, where the algorithm showed 86.8% accuracy in the laboratory and 85.2% accuracy outdoors. Dogan and Akcamete [14] performed an FFH detection study by calculating the fall height from three-axis acceleration data, with an overall error rate of 10.8%. Kim et al. [15] developed an FFH detection algorithm by calculating the vertical velocity and the trunk angles from IMU data and reported that 100% accuracy and a lead time of 301.8 ± 87.8 ms were obtained.

In order to increase survival rates after FFHs [16], it is important to predict the risk levels of falls. Arena et al. [17] experimentally confirmed that the peak acceleration of the head is between 4 and 11 m/s^2^ during falls. The peak acceleration value is one of the key measurement factors that can affect the severity of injury [18]. Kim et al. [19] proposed a study to predict the impact of falls of the elderly with the peak acceleration value. A regression analysis was performed using a deep learning algorithm based on IMU sensor data, and its results showed a mean absolute percent error of 6.69 ± 0.33% and an *r* value of 0.93. The risk of FFH accidents can be represented using the peak acceleration value. Risk prediction is more necessary and challenging to discriminate since FFH accidents result in more fatal injuries.

In this study, an IMU sensor was attached to the subject to obtain data regarding frequently observed or dangerous behaviors at construction sites. A human dummy was used to acquire data for falls from 2 m or above. Regression analysis was performed using various deep learning models (1D-CNN, 2D-CNN, LSTM, and Conv-LSTM) applied to the three-axis acceleration, three-axis angular velocity, and their SVM feature vectors. The risk level was estimated based on the predicted peak acceleration.

## 2. Materials and Methods

### 2.1. Experiment

A total of 20 healthy adult males (24.8 ± 2.0 years old, 173.5 ± 6.1 cm, 76.6 ± 13.0 kg) were recruited from Yonsei University for the study. Participants who had musculoskeletal problems were excluded. The experiment was conducted with the approval of the Yonsei University Mirae Campus IRB (1041849-202004-BM-042-02), and written consent was obtained from the participants [20].

An IMU module based on MPU-9250 (InvenSens, San Jose, CA, USA) was attached to the T7 position of the subject. The three-axis acceleration and angular velocity signals were measured at a sampling frequency of 100 Hz. LabVIEW (National Instruments, Austin, TX, USA) was used to save the data on a PC.

Fifteen general working movements (NF: non-fall), five low-hazard fall movements, (LF), and five high-hazard FFH movements (HF) were selected based on the reports of safety and health [8,21,22] and measured three times. For safety reasons, a dummy (Madamade, Chuncheon, Gangwon-do, Korea) (height: 180 cm, weight: 10 kg) was used for the HF experiments with fall heights higher than 2 m. Vertical and forward falling movements (FFH) were performed by the dummy at heights of 2 m and 3 m. All movements were repeated five times. Our previous study [15] revealed no differences between the data obtained when a person fell forward from 0.7 m and when a dummy fell forward from the same height using an SPSS-based independent sample *t*-test.

Table 1 represents the experimental movements in this study, comprising NFs, LFs, and HFs. Four sets of HF movements (HF01, 02, 04, and 05) were performed using the dummy, while HF03 was performed by participants. For all subjects, the experiments were performed based on the videos that were made on the Internet.

### 2.2. Pre-Processing

For the training, 70% of the human data and 60% of the dummy data were used (30% of the human data and 40% of the dummy data were used for the testing). Two sum vector magnitude (SVM) values, acceleration SVM (*A_SVM_*) and gyro SVM (*G_SVM_*), were calculated from the acceleration and angular velocity data measured by the IMU sensor [19]. Eight features were used to train the deep learning models, which predicted the magnitude of the impact acceleration. (Table 2). The data from 0.7 s to 0.2 s prior to the peak *A_SVM_* value were extracted for the analysis (Figure 1). The data analysis was conducted on a desktop equipped with Intel i7-12700 2.1 GHz, 32 GB RAM, NVIDIA GeForce RTX3060, and Windows 11.


(1)
$$A_{SVM}=\sqrt{Acc_{x}^{2}+Acc_{y}^{2}+Acc_{z}^{2}}$$



(2)
$$G_{SVM}=\sqrt{Gyro_{x}^{2}+Gyro_{y}^{2}+Gyro_{z}^{2}}$$


### 2.3. Deep Learning Models

Python 3.9 (Python Software Foundation, Wilmington, DE, USA) and the TensorFlow 2.9.0 library were used for deep learning. One-dimensional CNNs are mostly used for 1D time-series analyses [23], whereas 2D-CNNs are generally used for image classification or facial recognition in the form of 2D filters [24]. The long short-term memory (LSTM) method is a neural network model devised to supplement the drawback of a recurrent neural network (RNN), which is losing the first input information through multiple layers. Unlike RNN, LSTM introduces a structure where major information from the previous step is inputted into the next step [25]. Conv-LSTM utilizes the advantages of CNN and LSTM and is particularly useful for time-series prediction [26]. Data features are extracted from the convolutional layer of CNN, while LSTM receives the extracted features as input [27].

Figure 2 illustrates the structure of the four deep learning models. In Figure 2A,B, 1D-CNN and 2D-CNN models show two convolution layers in which overfitting is prevented by reducing the computation process through the max-pooling layer [28]. Figure 2C illustrates two layers of LSTM before the dropout layer. The Conv-LSTM comprised two convolution layers and two LSTM layers, in which the max-pooling layer was positioned after the convolution layer. Overfitting was prevented using the dropout (0.25) layer before the output. The basic hyperparameters for the four models were as follows: The Conv1D layer had a kernel size of 2 and a stride of 2. The max-pooling 1D and max-pooling 2D layers were 2 and (2, 2), respectively. Conv2D had a kernel size of (4, 4) and a stride of (2, 2). The activation function was ReLU, while the batch size of each model was 1. In addition, an early stop function was used to prevent overfitting. The models with an LSTM layer were set to include 50 epochs with 10 as patience. For the two CNNs, the epoch and the patience were set to 200 and 100, respectively.

The number of convolution filters and LSTM memory units were optimized using a grid search method. The numbers of filters in the first and the second layers for both the 1D-CNN and 2D-CNN were set to 8, 16, 32, and 64, and the numbers of units in the LSTM were 8, 16, 32, and 64. The numbers of filters in the conv1D layer of the Conv-LSTM were 16 and 64, while the numbers of units in the LSTM layer were 16 and 64.

### 2.4. Evaluation Methods

The errors between the measured and predicted values were calculated using two error functions: the mean absolute error (MAE) and the mean squared error (MSE).
(3)$$MAE=\frac{1}{N}\sum_{i=1}^{N}\left|\hat{y_{i}}-y_{i}\right|$$
(4)$$MSE=\frac{1}{N}\sum_{i=1}^{N}{\left(\hat{y_{i}}-y_{i}\right)}^{2}$$
where $\hat{y_{i}}$ is the peak acceleration value predicted by the model, and $y_{i}$ is the peak acceleration value measured in the experiment. The units of *MAE* and *MSE* are g and g^2^ respectively.

The accuracy, sensitivity, and specificity were calculated as
(5)$$Sensitivity\;\left(\%\right)=\frac{True\;Positives}{True\;positives+False\;negatives}\times 100,$$
(6)$$Specificity\;\left(\%\right)=\frac{True\;negatives}{True\;negatives+False\;positives}\times 100,$$
(7)$$Accurracy\;\left(\%\right)=\frac{True\;Positives+True\;negatives}{True\;positives+True\;negatives+False\;positives+False\;negatives}\times 100,$$
where *True positives* is the number of FFHs detected as FFHs, *False positives* is the number of non-FFHs detected as FFHs, *True negatives* is the number of non-FFHs detected as non-FFHs, and *False negatives* is the number of FFHs detected as non-FFHs.

## 3. Results

Table 3 represents the best performance among the models trained with hyperparameters. The smallest error (MAE: 1.46 g) was observed when the 1D-CNN model was trained with the MAE and the number of filters in the first and the second layers was 16. The error was the smallest (MSE = 6.02 g^2^) when the 1D-CNN model was trained with the MSE and the numbers of filters in the first and the second layers were 64 and 8, respectively. The smallest error (MAE: 1.61 g) was observed when the 2D-CNN model was trained with the MAE and the numbers of filters in the first and the second layers were 32 and 16, respectively. The error was smallest (MSE = 9.51 g^2^) when the 2D-CNN model was trained with the MSE and the numbers of filters in the first and the second layers were 32 and 16, respectively. The smallest error (MAE: 2.07 g) was observed when the LSTM model was trained with the MAE and the numbers of units in the first and the second layers were 8 and 16, respectively. On the other hand, the error was smallest (MSE = 12.20 g^2^) when the LSTM model was trained with the MSE and the numbers of filters in the first and the second layers were 8 and 64, respectively. The smallest error (MAE: 1.36 g) was observed when the Conv-LSTM model was trained with the MAE and the numbers of filters in the first and the second layers were 64 and 64, respectively and the numbers of units in the third and the fourth layers were 64 and 16, respectively. The error was smallest (MSE = 5.69 g^2^) when the Conv-LSTM model was trained with the MSE and the numbers of filters in the first and the second layers were 64 and 16, respectively and the numbers of units in the third and the fourth layers were 16 and 64, respectively.

Figure 3 shows the means and the standard deviations of the true and the predicted values for all experimental movements. The NF movements, except for NF15 (jumping), showed true values between 1 g and 4 g (5 g~9 g for NF15). The 2D-CNN and LSTM predicted peak acceleration values of less than 10 g in NF15, but both the 1D-CNN and Conv-LSTM trained by the MSE predicted values higher than 10 g. The LF movements were within the range of 5 g~13 g, whereas most HF movements had values higher than 9 g. The LF movements, especially LF03 and LF04, showed larger deviations than the other NF or HF movements. The LF03 and LF04 movements had peak acceleration values higher than 10 g, while the other LF movements were in the range of 4 g~9 g. Most predicted values in the LF were less than 9 g. The 1D-CNN, 2D-CNN, and LSTM (except Conv-LSTM) had significantly underestimated predictions, although at least 9 g of the peak acceleration values was measured in HF01. The vertical FFHs (HF01 and HF02) revealed smaller peak acceleration values than the forward FFHs (HF04 and HF05). The 1D-CNN and Conv-LSTM predicted peak acceleration values of HF movements beyond 10 g when estimated by the MAE. It is noted that the 2D-CNN predicted peak acceleration values with the largest deviation. The 1D-CNN and Conv-LSTM predicted the peak values in the HF movements better than the others. Considering the predicted peak acceleration values in the LFs and HFs, movements higher than 9 g could be defined as the threshold of FFH.

Table 4 presents the classification performance, assuming that a peak acceleration of 9 g or higher indicates FFH movements. The 1D-CNN showed better sensitivity (MAE: 83.3%, MSE: 87.5%) than the 2D-CNN (MAE: 4.2%, MSE: 79.2%), even though more than 90% accuracy was found in the 1D-CNN (MAE: 92%, MSE: 93.9%) and 2D-CNN (MAE: 90.7%, MSE: 96.5%). The LSTM demonstrated very poor sensitivity since its predicted values were mostly small. Among all the deep learning models, the Conv-LSTM showed the highest classification accuracy when trained by the MAE (97.6%).

## 4. Discussion

Our study shows that the 1D-CNN had smaller errors than the 2D-CNN. This implies that a 1D-CNN is more appropriate for time-series data. The Conv-LSTM showed the smallest errors among the others in both the MAE and MSE. The LSTM was used to predict the peak values, while the CNN was efficient in extracting features.

An NF15 (0.7 m jump) can frequently occur at construction sites; it is not necessarily a falling movement, but a dangerous one leading to an FFH. It showed larger predicted values of peak accelerations than the other NF movements; thus, it can be classified as an FFH. LF movements can easily occur at construction sites and are mostly predicted within the range of 4 g~9 g, for which the 1D-CNN and Conv-LSTM models obtained reasonable predictions. Such movements might not be too dangerous for the workers to require a life-saving wearable airbag, but they still might lead to serious injuries for the elderly during daily life.

In Figure 3, HF01 and HF02, vertical FFHs, showed relatively smaller peak acceleration values than HF03, HF04, and HF05, representing forward FFHs. In vertical FFHs, the foot first touches the ground during falling, which results in a smaller transfer of mechanical energy and, therefore, a smaller impact compared to forward FFHs. In Figure 3, the 1D-CNN and the Conv-LSTM predicted HFs more accurately, showing the high peak acceleration values, than the 2D-CNN and LSTM. The 2D-CNN and LSTM showed under-estimated peak acceleration values since a data imbalance appeared between a large number of NFs and a small number of HFs. This can be solved using data augmentation techniques, but they have overfitting problems.

As for estimating errors in prediction, the MSE resulted in smaller deviations between the sensitivity and the specificity of all deep learning models than the MAE. The MSE amplified errors by squaring, and the errors in the LFs and HFs were larger than those in the NFs. The MSE tended to focus on the LFs or HFs more than the NFs since deep learning models train to minimize errors. The 2D-CNN trained by the MAE had the poorest sensitivity (4.2%). The LSTM showed high accuracy (MAE: 94.4%, MSE: 92.0%), but significantly low sensitivity (MAE: 45.8%, MSE: 50.0%). The sensitivity is more important than the specificity since it is directly related to the notification of dangerous situations. The 1D-CNN and Conv-LSTM showed higher sensitivity than the others. Thus, 1D-CNN- or Conv-LSTM-based risk prediction algorithms would be more effective for safety issues. A risk prediction algorithm can be used as a device to warn of risks and call for emergency support when applied to a mobile application. In addition, the classification of NFs and LFs from HFs may be helpful when used with protective equipment such as an airbag. The aforementioned can lead to a reduction in the death rate.

There are some limitations to the study. First, our dataset does not represent all movements at construction sites since it was obtained from the simulated movements of only twenty participants. However, it is enough to represent major movements at construction sites and sufficient for developing an algorithm for predicting the risk of movements. Second, the developed algorithm tended to underestimate the risk of each movement, since the amount of data in each class (NF, LF, and HF) was unbalanced. Several augmentation techniques can be applied to solve this problem in the future.

## 5. Conclusions

The risks of FFHs at construction sites were predicted using IMU sensor data for accident prevention. A total of 15 non-falls (NF), 5 low-hazard falls (LF), and 5 high-hazard FFHs (HF) were selected as the experimental movements. Four deep learning models (1D-CNN, 2D-CNN, LSTM, and Conv-LSTM) and two error functions (MAE and MSE) were applied to predict the peak acceleration values at impact. The Conv-LSTM trained by the MAE showed the smallest error (1.36 g). When the threshold of 9 g was applied as the peak acceleration, the Conv-LSTM showed the highest accuracy (97.6%). Our algorithm successfully predicted the risk of movements and can be applied to real construction sites in the future. This study could be very helpful to prevent fatal injuries at construction sites by providing not only proper feedback to avoid unsafe behaviors but can also prompt treatment after accidents.

## Figures and Tables

**Figure 1 sensors-22-06107-f001:**
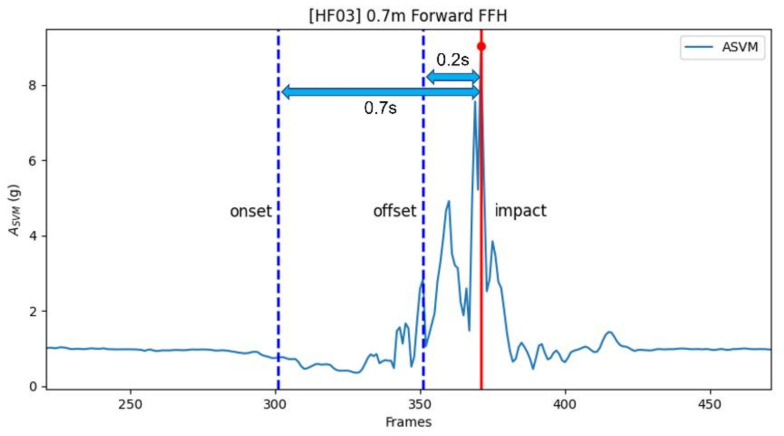
An example of the window extraction.

**Figure 2 sensors-22-06107-f002:**
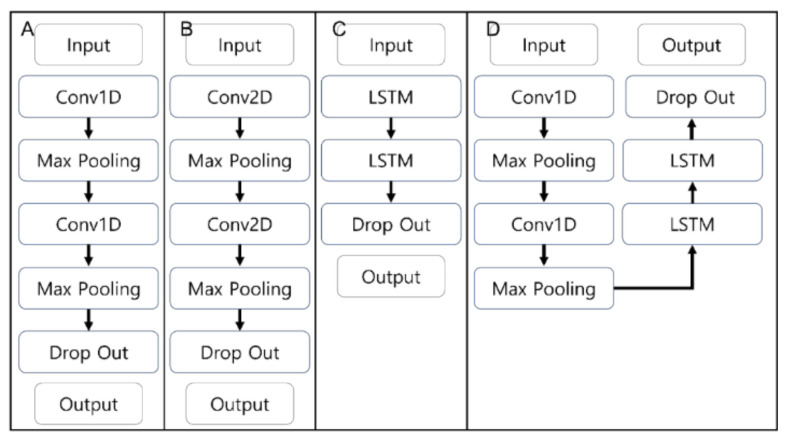
Structures of deep learning models: (**A**) 1D-CNN, (**B**) 2D-CNN, (**C**) LSTM, and (**D**) Conv-LSTM.

**Figure 3 sensors-22-06107-f003:**
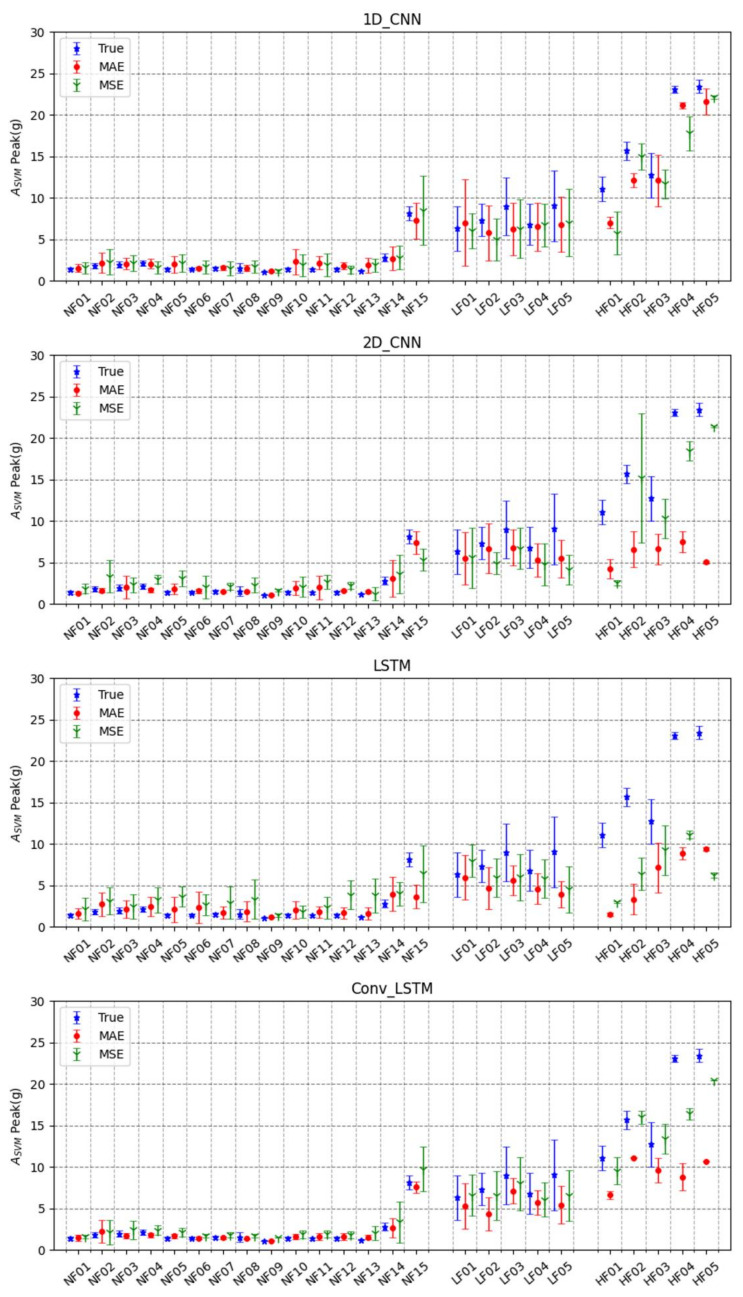
True vs. predicted (MSE, MAE) values of trained models for different movements.

**Table 1 sensors-22-06107-t001:** Experimental movements.

Non-Fall (NF)	NF01	Sitting quickly and getting up	NF09	Moving up and down in an elevator
	NF02	Sitting and getting up comfortably	NF10	Walking on a beam
	NF03	Going up and down the stairs	NF11	Walking on a beam with luggage
	NF04	Going up and down a ladder	NF12	Shoveling
	NF05	Working with a pickaxe	NF13	Stretching
	NF06	Lifting (front)	NF14	Climbing up and down a scaffold
	NF07	Lifting (back)	NF15	0.7 m jump
	NF08	Lifting (side)		
Low-Hazard Fall (LF)	LF01	Forward trip	LF04	Backward slip
	LF02	Lateral trip	LF05	Fainting
	LF03	Forward slip		
High-Hazard FFH (HF)	HF01	2 m Vertical FFH	HF04	2 m Forward FFH
	HF02	3 m Vertical FFH	HF05	3 m Forward FFH
	HF03	0.7 m Forward FFH		

**Table 2 sensors-22-06107-t002:** Eight features used in this study.

No.	Feature	No.	Feature
1	$A_{X}:\;$x-axis acceleration	5	$G_{X}$: x-axis angular velocity
2	$A_{Y}:\;$y-axis acceleration	6	$G_{Y}$: y-axis angular velocity
3	$A_{Z}:\;$z-axis acceleration	7	$G_{Z}:\;$z-axis angular velocity
4	$A_{SVM}$: Sum vector magnitude of acceleration	8	$G_{SVM}$: Sum vector magnitude of angular velocity

**Table 3 sensors-22-06107-t003:** Best performances of deep learning models (in terms of MAE and MSE).

Model Name	MAE (Epoch)	MSE (Epoch)
1D-CNN	1.46 g (183)	6.02 g^2^ (151)
2D-CNN	1.61 g (130)	9.51 g^2^ (187)
LSTM	2.07 g (18)	12.20 g^2^ (13)
Conv-LSTM	1.36 g (25)	5.69 g^2^ (49)

It is noted that the 1D-CNN showed smaller errors in prediction than the 2D-CNN in both the MAE and MSE. Among four deep learning models, Conv-LSTM demonstrated the best prediction results, and LSTM the poorest.

**Table 4 sensors-22-06107-t004:** Classification performances.

	Model							
	1D-CNN		2D-CNN		LSTM		Conv-LSTM	
Error Function	MAE	MSE	MAE	MSE	MAE	MSE	MAE	MSE
Accuracy (%)	92.0	93.9	90.7	96.5	94.4	92.0	97.6	92.3
Sensitivity (%)	83.3	87.5	4.2	79.2	45.8	50.0	62.5	95.8
Specificity (%)	92.6	94.3	96.6	97.7	97.7	94.9	100	92.0

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
