# Peer review of "Fall-from-Height Detection Using Deep Learning Based on IMU Sensor Data for Accident Prevention at Construction Sites"

_sensors, 2022, doi:10.3390/s22166107_

Round 1

Reviewer 1 Report

Thank you for the opportunity to review this article, below are my thoughts on the manuscript.

Summary:

- In the abstract it is not clear what the objective of the study is, nor what the study design is, the authors must make these aspects clear in the abstract. In the abstract the authors mention that the study was conducted to estimate the risk of falls from a height, however, at the end of the abstract the risk is not mentioned, I suggest rewriting the abstract with this informations highlighted.

Methods:

- What is the study design?

- How did the authors arrive at this sample size? Was there a calculation to estimate the sample size? Or is it a convenience sample?

- What were the eligibility criteria for volunteers to participate in the study?

- How were the volunteers recruited? Where were they recruited from?

- Does this sample physically represent construction workers?

- Why did the authors not carry out the study with the professionals in the area themselves?

- The authors used a dummy with a dummy weight of 10 kg, which does not represent the body weight of a construction worker. So I question the authors could this have influenced the results of this study?

Discussion:

- The authors justified carrying out this study in the introduction with data related to accidents and occupational health, which did not occur in the discussion.

- The discussion is the poorest part of this article and I missed discussing the findings with the real life of the workers.

- Would it be interesting to invest in this equipment to prevent these workers from falling? It's unclear why it would be interesting invest in this equipament in this version of the manuscript.

- I missed the authors mentioning the limitations of the study and still of a conclusion more focused on occupational health and not just with sensor data.

Author Response

Thanks for your comments for our  manuscript. Based on your comments, the manuscript was revised. Response for each comment was attached in the file. 

Reviewer 2 Report

The topic of this study is in the scope of the journal.

English language is appropriate throughout the manuscript, which is logical structured.

I am not sure, if the body height and body weight is important for the abstract, particular these are values from the population of this study. Please provide information on participants in abstract.

The introduction provides a good, generalized background of the topic that quickly gives the reader an appreciation of the wide range of applications for this technology.

The objective is clearly defined in the last sentence introduction.

The experimental apparatus is quite good, and is appropriate for the study, especially given that

the main focus of the paper is to develop a novel technique, but to demonstrate the power of that technique in scientific manner.

I think the discussion section of this study need to be made clearer. In particular, the connection between results and application in field and real situation shod be provide. As suggested above, I think a more in-depth discussion would be helpful. I feel this is an important for this paper, and therefore it merits more discussion. Can the author demonstrate why this results are helpful to prevent fatal injuries in construction sites?

No significant limitations are discussed.

Author Response

(The authors gave the same response as above.)

Round 2

Reviewer 1 Report

After the changes in the text, the article became clearer for the readers and with these changes I believe that the article is ready for publication.

Reviewer 2 Report

Dear Authors,

Thank you for precious time and invaluable comments and new version of manuscript. This paper has a potential to be accepted. This is an interesting study and the authors have collected a unique and interesting dataset using cutting edge methodology. The paper is generally well written and structured.